# Endometrium as Control of Endometriosis in Experimental Research: Assessment of Sample Suitability

**DOI:** 10.3390/diagnostics12040970

**Published:** 2022-04-12

**Authors:** Vince Szegeczki, László Fazekas, Máté Kulcsár, Dora Reglodi, Péter Török, Brigitta Orlik, Antonio Simone Laganà, Attila Jakab, Tamas Juhasz

**Affiliations:** 1Department of Anatomy, Histology and Embryology, Faculty of Medicine, University of Debrecen, Nagyerdei krt. 98, H-4032 Debrecen, Hungary; szegeczki.vince@anat.med.unideb.hu (V.S.); fazekasl33@gmail.com (L.F.); k.mate2010@gmail.com (M.K.); 2Department of Anatomy, PTE-MTA PACAP Research Team, Szentagothai Research Center, Medical School, University of Pécs, Szigeti út 12, H-7624 Pécs, Hungary; dora.reglodi@aok.pte.hu; 3Department of Obstetrics and Gynecology, Faculty of Medicine, University of Debrecen, Egyetem tér 1, H-4032 Debrecen, Hungary; torok.peter@med.unideb.hu (P.T.); ja@med.unideb.hu (A.J.); 4Department of Pathology, Faculty of Medicine, University of Debrecen, Egyetem tér 1, H-4032 Debrecen, Hungary; orlik.brigitta@med.unideb.hu; 5Unit of Gynecologic Oncology, ARNAS “Civico–Di Cristina–Benfratelli”, Department of Health Promotion, Mother and Child Care, Internal Medicine and Medical Specialties (PROMISE), University of Palermo, 90127 Palermo, Italy; antoniosimone.lagana@unipa.it

**Keywords:** endometrium, curettage, hysteroscopy, endometrial sampling, in vitro endometrial culturing

## Abstract

Endometriosis is a chronic gynecological disease that causes numerous severe symptoms in affected women. Revealing alterations of the molecular processes in ectopic endometrial tissue is the current policy for understanding the pathomechanisms and discovering potential novel therapeutic targets. Examining molecular processes of eutopic endometrium is likely to be a convenient method to compare it with the molecular alterations observed in ectopic tissues. The aim of the present study was to determine what proportion of the surgically resected eutopic endometrial samples is suitable for further experiments so that these can be comparable with endometriosis. Final hospital reports and histopathology reports of a 3-year-long period (1162 cases) were analysed. The application of a retrospective screening method promoted the categorization of these cases, and quantification of the categorized cases was accomplished. In addition, results obtained from cultured endometrium samples were also detailed. Only a small number of the harvested endometrial samples was suitable for further molecular analysis, while preoperative screening protocol could enlarge this fraction. Applying clinical and histopathological selection and exclusion criteria for tissue screening and histopathological examination of samples could ensure the comparability of healthy endometrium with endometriosis. The present study could be useful for researchers who intend to perform molecular experiments to compare endometriosis with the physiological processes of the endometrium.

## 1. Introduction

Endometriosis is a gynaecological disease described as the formation of endometrial-like tissue outside the uterine cavity [1]. Although the definite origin and pathogenesis of the disease are still unknown [2], several theories exist describing the development of the lesions [3]. In addition to the widely accepted retrograde menstruation theory, embryonic rest, lympho-vascular metastasis, and coelomic metaplasia hypotheses have also been settled in the last decade [4]. In recent studies, stem cell-originated endometriosis has also been suggested [5]: ectopic tissue occurs from endogenous stem cells in the endometrium or from bone-marrow-derived stem cells that differentiate into endometrial cells [6]. Various molecular alterations have been described in endometriosis that may affect several emerging downstream pathways such as transcriptional regulation, cell cycle regulation, and cell adhesion [7,8].

The endometrium is a multicellular tissue that forms the inner layer of the uterus. Different processes that lead to the development of endometriosis can result from physiological alterations observed in the endometrium [8]. Hence, it may be hypothesized that eutopic endometrium can be an appropriate control to detect the different molecular changes observed in the ectopic tissue. However, due to many reasons, it is not easy to harvest healthy human endometrium, and careful consideration is needed to reveal the different molecular processes of this tissue [9]. Therefore, the aim of this present study is to assess the suitability of the clinically extracted endometrial samples for further experimental research. To answer this question, more than 1000 cases of a 3-year-long period were examined, where endometrial harvesting happened. In addition, with the help of eight cultured samples, examples of possible general pitfalls during the selection of suitable samples are shown.

## 2. Materials and Methods

### 2.1. Collection of Data

Lists from a medical database of patients provided by the Medical Record Office (University of Debrecen) were studied (Research ethics committee approval number: H.0180-2020). Cases based on this list were collected from the period between January 2017 and March 2020, when samples of endometrial biopsies and curettages were extracted by the surgeons of the Obstetrics and Gynaecology Clinic. These tissues were submitted with an ICD (international classification of diseases) code of irregular menstrual bleeding to the Pathology Department for histopathological analysis. In total, the final hospital reports and histopathology reports of 1162 cases were analysed. The following entries were collected: the age of the patient, the appropriate surgical indication, the type of surgery, and finally, the diagnosis in the histopathology report.

### 2.2. Grouping of Histopathological Diagnoses

In total, 21 different diagnoses were described in the histopathology reports. Firstly, all cases were separated into these 21 groups. To make it easier to handle the different cases, all diagnoses were categorized into 10 groups based on their similarities. These were the following: proliferative phase endometrium, secretory phase endometrium, menstrual phase endometrium, group for effects of exogenous hormones, menopausal endometrium, samples inadequate for analysis, endometrial polyp, endometrial hyperplasia, malignant tumors, and endometritis (Table 1).

### 2.3. Using Clinical Exclusion Criteria

In freshly extracted samples, the diagnosis included in the histopathology report was unknown at the time the tissue was submitted to a laboratory for in vitro tissue culturing. Therefore, all samples suggesting underlying pathological processes of the uterus must be eliminated. Every case suitable for further molecular analysis was selected. The following aspects constituted the basis of exclusion: 1. every sample was excluded in which the patient was older than 45 years; 2. surgery was performed because of an established pathological process of the uterus or the type of operation was not only aimed at the extraction of the endometrium; 3. endometriosis or pathological conditions can be revealed in the anamnesis of the patient. The following cases of surgical indication were excluded: postmenopausal bleeding, perimenopausal bleeding, cervical atypia, IUD removal, pathological findings on ultrasound, and preoperative curettage before excision of an intrauterine tumor (Table 2).

### 2.4. Using Clinical Selection Criteria

Considering the exclusion criteria, every case was selected where the patient was younger than 45 years, where surgery was performed because of heavy menstrual bleeding (menorrhagia, abnormal menstruation, irregular periods), and finally, where an operation was aimed at the extraction of the eutopic endometrium. The following types of operation were selected: D&C (dilation and curettage) scraping procedures, such as curettage, fractional curettage (F&C), abrasion and fractional abrasion, in addition to hysteroscopy with endometrial biopsy (HSC) and transcervical resection of endometrium (TCRE). Scraping (D&C) and hysteroscopy with endometrial biopsy (HSC) procedures were also examined separately from each other (Table 2).

### 2.5. Using Histopathological Selection and Exclusion Criteria

Knowing the diagnoses of the histopathology report, only proliferative and secretory phase endometria were accepted as appropriate control samples for experimental research. Considering these diagnoses only, every case was excluded where the histopathology report described a condition that could affect the signalling pathways of the physiological endometrium: disordered proliferative endometrium, adenomyosis, and cervical intraepithelial neoplasia (CIN) (Table 2).

### 2.6. Tissue Culturing

Endometrial tissues harvested by biopsy during HSC procedures were obtained from the Department of Gynaecology. Tissues were processed for histopathological analysis with ICD codes different from irregular menstrual bleeding. Evaluation of the uterine cavity was performed as part of diagnostic hysteroscopy. No other uterine pathologies could be detected preoperatively. Every patient was younger than 45 years. These samples were minced into 60 × 15 mm cell culture dishes (Eppendorf, North America, Inc., New York, NY, USA) and were fixed to a 15 µL matrigel drop (Cultrex^®^ BME, Type 2). Dishes were filled with 4.5 g/L Glucose DMEM (Lonza, Bend, OR, USA), and the medium was changed every day. Tissues were divided into control and hormone-treated groups. Knowing the first day of the last menstruation (LMP), samples were maintained until the 24th day of the menstrual cycle. During the whole culturing period, protocol against tissue contamination was kept.

### 2.7. Hormone Application

For hormone administration, 17β-estradiol (E2) and progesterone (P4) solutions (Sigma-Aldrich, St. Louis, MO, USA) were applied to the medium of the hormone-treated groups. Given the first day of the LMP, whenever the endometrial sample was submitted to the laboratory, the relevant day of the menstrual cycle could be calculated. Four different concentrations of E2 and P4 solutions were available. This way, the final concentrations of hormones in the medium were the mean serum levels observed in the early follicular, follicular, ovulatory, and luteal phases of the ovarian cycle. Until the last day of hormone administration, hormone concentrations were changed when the sample reached another phase of the ovarian cycle (Table 3). The aim of this procedure was to imitate the in vivo hormonal changes of the menstrual cycle.

### 2.8. H&E Staining

After the last day of the treatment period, tissues were fixed in 4% paraformaldehyde (PFA) solution for 6 h at least. On the day of sample submission, one part of the removed tissue (‘Fresh’) was fixed directly after the operation. Following paraffinization, histological slides were sectioned to StarFrost^®^ microscope slides (Knittel Glass, Brunswick, Germany). After deparaffinization, slides were stained with haematoxylin (VWR International, Radnor, PA, USA) and eosin (Amresco, Fountain Parkway Solon, OH, USA) histological staining method. Photomicrographs with 10×, 20× and 40× magnifications were taken from the stained slides with a light microscope (BX-53 Microscope, Olympus Microscopes).

### 2.9. Mathematical Analysis

After counting the number of cases in different histopathological diagnostic groups during the retrospective analysis, data were converted to percentages to make the results more transparent.

### 2.10. Ethical Approval

Research ethics committee approval numbers of the present study are the followings: H.0180-2020 and 28966-2/2018/EKU.

## 3. Results

### 3.1. Histopathological Findings of the Samples

In the histopathology reports, 17.64 and 10.67% of the samples were diagnosed as proliferative and secretory phase endometria. The remaining tissues were verified as menstrual phase endometria (1.46%), menopausal endometria (4.56%), and pathological endometria: endometrial polyp (13.51%), endometrial hyperplasia (24.01%), malignant tumors (4.04%), and endometritis (1.72%). Overall, 16.18% of the samples seemed inadequate for further analysis, and 6.20% of the endometria showed signs of exogenous hormonal effects (Table 4). All histopathological diagnoses of these mentioned 10 groups are detailed in Table 1.

### 3.2. Histopathological Findings of Endometrial Scraping and Biopsy Procedures

Of the total 1162 cases aiming at the extraction of the endometrium, 949 tissues were removed by only scraping and hysteroscopy with endometrial biopsy surgical procedures. The resection of the remaining 213 samples was an additional surgery or another surgical procedure. Knowing the histopathology reports of these 949 cases, 15.07% and 10.01% of the samples were diagnosed as proliferative and secretory phase endometria. The remaining tissues were verified as menstrual phase endometria (1.79%), menopausal endometria (4.32%), and pathological endometria: endometrial polyp (12.33%), endometrial hyperplasia (26.66%), malignant tumors (4.43%), and endometritis (1.05%). Overall, 19.07% of the samples seemed inadequate for further analysis, and 5.27% of the endometria showed signs of exogenous hormonal effects (Table 5). All histopathological diagnoses of these mentioned 10 groups are detailed in Table 1.

### 3.3. Histopathological Findings of Endometrial Scraping Procedures (D&C)

From the previously mentioned 949 cases, 833 endometria were extracted by scraping surgical procedures. In the histopathology reports of these 833 cases, 14.17 and 9.84% of the samples were diagnosed as proliferative and secretory phase endometria. The remaining tissues were verified as menstrual phase endometria (1.80%), menopausal endometria (4.44%) and pathological endometria: endometrial polyp (12.24%), endometrial hyperplasia (27.25%), malignant tumors (4.32%), and endometritis (1.20%). Overall, 19.69% of the samples seemed inadequate for further analysis, and 5.04% of the endometria showed signs of exogenous hormonal effects (Table 6). All histopathological diagnoses of these mentioned 10 groups are detailed in Table 1.

### 3.4. Histopathological Findings of Hysteroscopy with Endometrial Biopsy Procedures (HSC)

From the previously mentioned 949 cases, 116 endometria were extracted by hysteroscopy with endometrial biopsy procedures. In the histopathology reports of these 116 cases, 21.55 and 11.21% of the samples were diagnosed as proliferative and secretory phase endometria. The remaining tissues were verified as menstrual phase endometria (1.72%), menopausal endometria (3.45%), and pathological endometria: endometrial polyp (12.93%), endometrial hyperplasia (22.41%) and malignant tumors (5.17%). There were no histopathological findings for endometritis. Overall, 14.66% of the samples seemed inadequate for further analysis, and 6.90% of the endometria showed signs of exogenous hormonal effects (Table 7). All histopathological diagnoses of these mentioned 10 groups are detailed in Table 1.

### 3.5. Clinically Suitable Samples for In Vitro Experimental Research Extracted by Scraping and Hysteroscopy with Endometrial Biopsy Procedures

Without knowing the histopathological findings of the samples, only using the previously described clinical exclusion and selection criteria, altogether 145 endometria (15.28%) seemed suitable for further experimental analysis from the total 949 cases (Table 8). By retrospective analysis, it was proved that from these 145 tissues, 22.07% and 15.86% of the samples were diagnosed as proliferative and secretory phase endometria. The remaining tissues were verified as menstrual phase endometria (3.45%), menopausal endometria (1.38%), and pathological endometria: endometrial polyp (8.97%), endometrial hyperplasia (31.03%), malignant tumors (0.69%), and endometritis (0.69%). Overall, 10.34% of the samples seemed inadequate for further analysis, and 5.52% of the endometria showed signs of exogenous hormonal effects (Table 9). All histopathological diagnoses of these mentioned 10 groups are detailed in Table 1.

### 3.6. Clinically Suitable Samples for In Vitro Experimental Research Extracted by Scraping Procedures (D&C)

From the previously mentioned 145 cases, 116 endometria were extracted by scraping surgical procedures (Table 8). By retrospective analysis, only regarding these cases, it was proven that 19.83% and 17.24% of the samples were diagnosed as proliferative and secretory phase endometria. The remaining tissues were verified as menstrual phase endometria (3.45%), menopausal endometria (1.72%), and pathological endometria: endometrial polyp (9.48%), endometrial hyperplasia (31.90%), and endometritis (0.86%). There were no histopathological findings for malignant tumors. Overall, 10.34% of the samples were inadequate for further analysis, and 5.17% of the endometria showed signs of exogenous hormonal effects (Table 10). All histopathological diagnoses of these mentioned 10 groups are detailed in Table 1.

### 3.7. Clinically Suitable Samples for In Vitro Experimental Research Extracted by Hysteroscopy with Endometrial Biopsy Procedures (HSC)

From the previously mentioned 145 cases, 29 endometria were extracted by hysteroscopy with endometrial biopsy procedures (Table 8). By retrospective analysis, only regarding these cases, it was proven that 31.03 and 10.34% of the samples were diagnosed as proliferative and secretory phase endometria. The remaining tissues were verified as menstrual phase endometria (3.45%) and pathological endometria: endometrial polyp (6.90%), endometrial hyperplasia (27.59%), and malignant tumors (3.45%). There were no histopathological findings for endometritis or menopausal endometria. Overall, 10.34% of the samples were inadequate for further analysis, and 6.90% of the endometria showed signs of exogenous hormonal effects (Table 11). All histopathological diagnoses of these mentioned 10 groups are detailed in Table 1.

### 3.8. Suitable Samples for Further In Vitro Experimental Research Studies

For the reasons detailed below, only proliferative and secretory phase endometria are suitable for experimental research. Of the total number of examined samples (949 cases), 145 (15.3%) were suitable after using only clinical selection and exclusion criteria. From these, 116 were harvested by scraping and 29 by hysteroscopy with endometrial biopsy procedures. The number of samples suitable for molecular experiments (*suitability detailed* in Section 2.3, Section 2.4 and Section 2.5) was revealed with the help of retrospective analysis. Altogether, 32 samples were suitable by using histopathological selection and exclusion criteria. This is 3.37% of the 949 cases and 22.07% of the 145 cases. Regarding the scraping procedures, 28 endometria were suitable for further in vitro experimental analysis. This is 3.36% of the 833 and 24.14% of the 116 cases. Regarding the hysteroscopy with endometrial biopsy procedures, four endometria were suitable for further in vitro experimental analysis. This is 3.45% of the 116 and 13.79% of the 29 cases (Table 12).

### 3.9. Experiences Obtained from the Submitted and Cultured Eutopic Endometrial Tissues

Every sample was analysed with histopathological assessment. One piece of submitted tissue was fixed with 4% PFA on the day of endometrium removal to verify the histopathological diagnosis of the harvested sample, determine the actual histological status and be able to compare the submitted tissue with the cultured tissues. After the treatment period (following the 24th day of the menstrual cycle), the control and the hormone-administered samples were also assessed with histopathological analysis. Results are demonstrated with the example of different cases.

Case 1 endometrium (Appendix A) contained a thick layer of submucosal myometrium and was composed of secretory phase glands and stroma. However, the whole thickness of the secretory phase endometrium could not be observed since the layer close to the uterine cavity was missing. Furthermore, histopathological analysis of another sample harvested from the same uterus verified polypoid adenomyoma. In addition, this kind of thick myometrium layer in a sample always raises the possibility of the presence of submucosal uterine leiomyoma. Hence, this endometrium would be excluded from further molecular examinations. Moreover, cultured tissues underwent necrosis during the treatment period.

Case 2 endometrium (Appendix A) comprised secretory phase tissue. Nevertheless, in ‘Fresh’ samples, interstitial bleeding was observed, suggesting injury of the tissue. In this case, the culture also underwent necrosis.

In Case 3 endometrium (Appendix A) glands showed the signs of artificial injury. In addition, histopathological analysis of another removed piece from the same endometrium verified endometrial polyp. Hence, this endometrium would be excluded from further molecular examinations. Cultured tissues showed signs of necrosis, and in control and hormone-treated groups, severe necrosis of stroma was observed. Moreover, one minced portion contained squamous metaplasia of the endocervix.

Case 4 endometrium (Appendix A) showed the signs of another type of artificial injury. Gland cells separated from the stromal cells and disintegrated from each other.

Cultured tissues of Case 5 endometrium (Appendix A) underwent necrosis during the treatment period. Histopathological analysis of the ‘Fresh’ sample verified that the tissue was disordered proliferative endometrium as proliferative and secretory phase glands could be found alongside each other.

In Case 6 endometrium (Appendix A), another type of disordered proliferative endometrium was confirmed. In this sample, the appearance of proliferative phase stroma was observed along with secretory phase glands.

Histopathological analysis of Case 7 endometrium (Appendix A) confirmed healthy, secretory phase endometrium. Cultured tissues could be maintained until the last day of the treatment period. In control groups, secretory phase glands produced mucus. The same condition was observed in hormone-administered groups, where very mild necrosis was observed around the intact endometrial glands.

Case 8 endometrium (Appendix A) was similar to Case 7 endometrium. In hormone-treated samples, the secretory phase of endometrium appeared more progressed compared to the control group.

## 4. Discussion

The formation of endometriosis can be the result of the involvement of stem cell populations that can be found in the human endometrium [10]. The issue regarding the differences in gene expression between the eutopic and ectopic tissues has been raised in some studies [11,12]. Considering these, it seems logical to conclude that eutopic endometrium is a suitable candidate for comparison with endometriosis in experimental research. This way, changes detected in the ectopic tissue can be confirmed with the results obtained from a healthy endometrium. It could be a question of interest why it is inevitable to examine human endometrium in an in vitro experimental research instead of using endometrial cell lines or animal models.

Endometrial cells become terminally differentiated during each menstrual cycle [13], while endometrial stem cells (ESC) are responsible for the monthly regeneration of the endometrium [14]. These could be endometrial mesenchymal stem cells, endometrial epithelial progenitors, side population cells, and bone marrow-derived stem cells that support the proliferation of stromal and epithelial compartments and are located in both layers of the endometrium [15,16]. Basal and functional layers contain endometrial glands embedded in a multicellular stroma consisting of fibroblasts, immune cells and vascular components [17]. Regarding all these, a diverse population of cells can be found in the endometrium; they affect each other and form an endometrial milieu responsible for the complexity of the menstrual cycle and renewing the endometrial tissue. Nevertheless, endometrial cell lines used in different research are derived from pathological endometria or isolated from menstrual blood and generally comprise only one cell type (e.g., mesenchymal stem cells) of the endometrium [18,19,20,21]. Furthermore, these can also be immortalized cell lines [22]. Both the cell lines trying to model the processes of endometriosis [23] as well as the endometrial cell lines demonstrate one or several aspects of the physiology of the endometrium but are not applicable for understanding the entire complexity of the tissue. In addition, there is an absence of cell-cell interactions in these models, which are characteristic of the complex microenvironment of the endometrium [20].

One of the main problems with using animal models to investigate the processes of the endometrium is that most laboratory animals have an oestrus cycle instead of a menstrual cycle, except for menstruating primates [24,25]. The latter animals are expensive to house, and the use of them for routine screenings is unethical. [26]. Another controversy regarding the use of animals is that endometriosis only develops spontaneously in humans and menstruating primates [26]. Although several models have been used to examine the pathophysiology of the disease [27,28,29,30], they have limitations and can not mimic or reproduce all aspects of endometriosis [29,30].

As endometrial cell lines and animal models are not capable of fully reflecting the physiological characteristics of the endometrium [20], examining the human endometrium itself can be a convenient decision to reveal the physiological or pathological molecular processes of this tissue as they consist of the same cells. Histopathological diagnosis of endometriosis consists of identifying two or more of the following cell types: endometrial gland cells, endometrial epithelial cells, endometrial stromal cells and hemosiderin-laden macrophages [1]. The endometrial complexity and extensive capability of the tissue to transform are hallmarked by the diversity of cell types in the tissue, including epithelial, stromal, vascular (endothelium, pericytes and vascular smooth muscle), and immune cells [17,31,32]. Histologically, the endometrium is divided into a basal and a functional layer [33]. Because of the hormonal changes (different blood levels of oestrogen and progesterone), the endometrium undergoes cyclic episodes of proliferation, differentiation, and, in the absence of embryo implantation, the shedding of the functional layer [17]. Regeneration of endometrium after menstruation is provided by endometrial stem cells [34]. It has been found that these stem cells, found in both layers of the endometrium, can be distributed by retrograde menstrual efflux, and may contribute to the establishment of ectopic endometrium [35]. In recent studies, stem cell-originated endometriosis has also been suggested [5]: ectopic tissue occurs from endogenous stem cells in the endometrium or from bone-marrow-derived stem cells that differentiate into endometrial cells [6].

Harvesting healthy endometrium is a challenging process that also requires ethical consideration. Extraction of this sample from the uterine cavity is only possible by appropriate surgical indication. Moreover, operations are generally performed due to a presupposed underlying pathological process in the uterus. Three main surgical procedures provide the extraction of the endometrium: scraping (curettage), hysteroscopy, and hysterectomy procedures [9,36].

The aim of our present retrospective analysis was to determine what proportion of the surgically removed human eutopic samples are suitable as appropriate controls to reveal molecular alterations observed in ectopic tissues. This study covers the cases of a 3-year-long period and intends to reveal what the chances are that a healthy, surgically removed endometrial sample submitted to a laboratory is eligible to be processed for further in vitro experimental research. A list of 1162 final hospital reports and histopathology reports was examined where endometria have been submitted to histopathological analysis with an ICD code of irregular menstrual bleeding. This is one of the most commonly used codes submitted to the Pathology Departments together with endometrial samples. Altogether 949 operations were aimed at endometrial harvesting only. The remaining 213 cases were excluded, these being hysterectomy procedures or operations where endometrial harvesting was only a complementary part of another surgical procedure. In most cases, a hysterectomy is performed because of a confirmed pathological condition [9]. Hence, in the present research, only endometria extracted by scraping and biopsy procedures were examined. It has been described that in pathological endometria, numerous molecular alterations can be observed (e.g., in endometrial hyperplasia and tumors) [37,38,39,40]. For this reason, only healthy endometria can be an appropriate choice to examine molecular processes in a control experiment. In addition, it has been described that perimenopausal endometrium has altered gene expression compared to the premenopausal tissue [41]. Thus, as early menopause occurs at or before the age of 45 [42], every sample was excluded where the patient was older than 45 years.

At the time the endometrial tissue is submitted to an experimental research laboratory directly after the surgical procedure, histopathological diagnosis is unknown. In this case, based on our research, the chance of examining pathological endometrium is more than 96%. Hence, it should be considered to use preliminary clinical exclusion and selection criteria to decrease the probability of getting pathological tissue. With the help of these, it was shown that the chance of examining a healthy endometrium is more than 22%.

The reason for examining the D&C and HSC cases separately is that endometrial samples harvested with these procedures are examined in different types of molecular research. Tissues removed with scraping procedures mainly contain the functional layer and are suitable for fertility and perimenopausal research, while endometria harvested by hysteroscopy with endometrial biopsy also contain the basalis layer, and are suitable for stem cell examinations [9]. At the time of sample submission, the chance of examining physiological endometrium is a little more than 3% in the case of scraping and hysteroscopy with endometrial biopsy procedures also. With the help of using clinical exclusion and selection criteria, the chance of examining healthy endometrium is more than 24% regarding the scraping procedure and more than 13% for hysteroscopy with endometrial biopsy procedure. The proportion of endometrial samples with pathological findings was also revealed, although it was not the aim of the present research, contrary to previous studies [36,43].

To examine molecular processes of a healthy endometrium, the use of samples extracted with surgical indications other than menstrual disorders should be considered. Endometrium harvested during TCRS (transcervical resection of the uterine septum) or diagnostic hysteroscopy could be good candidate methods; nevertheless, histopathological findings observed in these kinds of samples have not been described yet. Removing the uterus of brain-dead women who are candidates for organ donation would also be a possibility to extract endometrium; however, it is supposed that these uteri would be used to treat absolute uterine factor infertility patients in the future [44]. The main limitation of the present research was that only samples submitted with an ICD code of irregular menstrual bleeding were examined, and endometria submitted with different ICD codes were not.

Regarding cultured samples, it was tested whether it is possible to maintain a minced piece of endometrium under laboratory circumstances. Hormonal changes in the menstrual cycle were mimicked with estradiol and progesterone administration. As a normal menstrual cycle lasts between 25 and 35 days [45], the length of the treatment period was 24 days and samples were fixed in formalin solution on the 25th day. Analysis of eight cases revealed the problems and difficulties that a molecular researcher faces while doing experiments with eutopic endometrium. First of all, submitted samples can be injured artificially at the time of removal. Harvested tissue can sustain mechanical injuries and interstitial bleedings during the operation. It is unlikely that these injuries are dependent on individual surgeons. Moreover, except for the layers of the endometrium, samples can be composed of other cell types, such as smooth muscle cells of submucosal myometrium or epithelial cells of the endocervix. These common pitfalls have also been described in previous studies [9]. Furthermore, in spite of applying clinical selection and exclusion criteria, during histopathological analysis, pathological conditions can be revealed, such as disordered proliferative endometrium or endometrial polyp. Regarding our results, healthy endometria outlive the culturing procedures and can be used for further analysis.

## 5. Conclusions

In summary, since the possibility of harvesting pathological tissue is very high, it is a major challenge for experimental researchers to achieve reliable results from human eutopic endometrium while examining molecular processes. Using clinical exclusion and selection criteria for screening pathological samples increases the chance of examining healthy endometrium. Retrospective histopathological analysis is indispensable to ensure that results were obtained from physiological tissue (Figure 1). Samples where the pathological condition was confirmed are recommended to be excluded from experimental research. This way, the results from eutopic endometrium can be compared with the molecular processes observed in endometriosis to reveal whether there are differences between the two tissues.

## Figures and Tables

**Figure 1 diagnostics-12-00970-f001:**
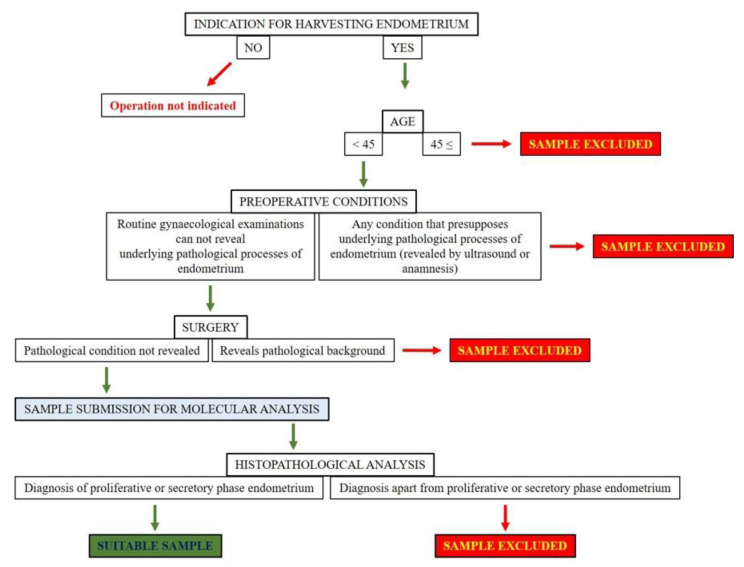
Recommended protocol for the involvement of eutopic endometrium samples to experimental research studies.

**Table 1 diagnostics-12-00970-t001:** 21 different histopathological diagnoses categorized into 10 groups.

Groups	Histopathological Diagnoses
Proliferative phase endometrium	- Proliferative phase endometrium
Secretory phase endometrium	- Secretory phase endometrium
Menstrual phase endometrium	- Menstrual phase endometrium
Exogenous hormones	- Pseudo-decidualization - Effects of gestagens
Menopausal endometrium	- Pseudo-menopausal endometrium - Endometrial atrophy
Inadequate for analysis	- Desquamated endometrium - Inadequate sample - Submucosal uterine leiomyoma
Endometrial polyp	- Endometrial polyp
Endometrial hyperplasia	- Simple endometrial hyperplasia - Simple glandular endometrial hyperplasia - Complex endometrial hyperplasia - Adenomatous endometrial hyperplasia - Simple glandular atypical endometrial hyperplasia - Complex atypical endometrial hyperplasia
Malignant tumors	- Endometrioid adenocarcinoma - Uterine carcinosarcoma
Endometritis	- Acute endometritis - Chronic endometritis

**Table 2 diagnostics-12-00970-t002:** Figure illustrating the applied selection and exclusion of cases with different examples.

Clinical Selection	Type of Operation	Indication of Surgery	Age	Histopathological Diagnosis	Endometriosis/Adenomyosis
	curettage	heavy menstrual bleeding (HMB)	46	proliferative phase endometrium	no
	LEEP + curettage	cervical cytologic atypia	28	CIN-II, proliferative phase endometrium	no
	curettage	heavy menstrual bleeding (HMB)	39	proliferative phase endometrium	no
	curettage	heavy menstrual bleeding (HMB)	44	disordered proliferative endometrium	no
	hysteroscopy (HSC)	pathological findings on ultrasound	44	proliferative phase endometrium	no
	hysteroscopy (HSC)	heavy menstrual bleeding (HMB)	26	proliferative phase endometrium	yes
	hysteroscopy (HSC)	heavy menstrual bleeding (HMB)	31	proliferative phase endometrium	no
	transcervical resection of polyp (TCRP)	pathological findings on ultrasound	28	disordered proliferative endometrium	no
	hysteroscopy (HSC)	pathological findings on ultrasound	39	proliferative phase endometrium	no
	curettage	pathological findings on ultrasound	48	proliferative phase endometrium	no
			selected		
			excluded		
			histopathologic exclusion		
			reason of exclusion		

**Table 3 diagnostics-12-00970-t003:** Hormone administration mimicking the hormonal changes of a 24 day long menstrual cycle. Mean serum levels of E2 and P4 were used for treatment.

Menstruation					Proliferative Phase									Secretory Phase									
1	2	3	4	5	6	7	8	9	10	11	12	13	14	15	16	17	18	19	20	21	22	23	24
***Early Follicaular Phase*** **E2** = 8.7–75 ng/L **→ 42 ng/L**																							
				***Follicular Phase*** **E2** = 12.5–166 ng/L → 89 **ng/L** **P4** = 0.06–0.9 µg/L → 0.5 **µg/L**																			
											***Ovulation*** **E2** = 85.8–498 ng/L **→** 292 **ng/L** **P4** = 0.1–12 µg/L **→** 6.1 **µg/L**												
																***Luteal Phase*** **E2** = 43.8–211 ng/L **→** 112.7 **ng/L** **P4** = 1.8–24 µg/L **→** 12.9 **µg/L**							

**Table 4 diagnostics-12-00970-t004:** Histopathological findings of 1162 cases.

Group	Case Number	Percentage (%)
Proliferative phase endometrium	205	17.64
Secretory phase endometrium	124	10.67
Menstrual phase endometrium	17	1.46
Exogenous hormones	72	6.20
Menopausal endometrium	53	4.56
Inadequate for analysis	188	16.18
Endometrial polyp	157	13.51
Endometrial hyperplasia	279	24.01
Malignant tumors	47	4.04
Endometritis	20	1.72
Total	**1162**	**100**

**Table 5 diagnostics-12-00970-t005:** Histopathological findings of cases where operation aimed at the extraction of endometrium.

Group	Case Number	Percentage (%)
Proliferative phase endometrium	143	15.07
Secretory phase endometrium	95	10.01
Menstrual phase endometrium	17	1.79
Exogenous hormones	50	5.27
Menopausal endometrium	41	4.32
Inadequate for analysis	181	19.07
Endometrial polyp	117	12.33
Endometrial hyperplasia	253	26.66
Malignant tumors	42	4.43
Endometritis	10	1.05
Total	**949**	**100**

**Table 6 diagnostics-12-00970-t006:** Histopathological findings of samples extracted by scraping procedure.

Group	Case Number	Percentage (%)
Proliferative phase endometrium	118	14.17
Secretory phase endometrium	82	9.84
Menstrual phase endometrium	15	1.80
Exogenous hormones	42	5.04
Menopausal endometrium	37	4.44
Inadequate for analysis	164	19.69
Endometrial polyp	102	12.24
Endometrial hyperplasia	227	27.25
Malignant tumors	36	4.32
Endometritis	10	1.20
Total	**883**	**100**

**Table 7 diagnostics-12-00970-t007:** Histopathological findings of samples extracted by hysteroscopy with endometrial biopsy procedure.

Group	Case Number	Percentage (%)
Proliferative phase endometrium	25	21.55
Secretory phase endometrium	13	11.21
Menstrual phase endometrium	2	1.72
Exogenous hormones	8	6.90
Menopausal endometrium	4	3.45
Inadequate for analysis	17	14.66
Endometrial polyp	15	12.93
Endometrial hyperplasia	26	22.41
Malignant tumors	6	5.17
Endometritis	0	0.00
Total	**116**	**100**

**Table 8 diagnostics-12-00970-t008:** Cases that were suitable by applying clinical selection and exclusion criteria.

Operation	Total Case Number	Clinically Suitable (Case Number)	Clinically Suitable (%)
D&C + HSC	949	145	15.28
D&C	833	116	13.93
HSC	116	29	25.00

**Table 9 diagnostics-12-00970-t009:** Histopathological findings of clinically suitable samples.

Group	Case Number	Percentage (%)
Proliferative phase endometrium	32	22.07
Secretory phase endometrium	23	15.86
Menstrual phase endometrium	5	3.45
Exogenous hormones	8	5.52
Menopausal endometrium	2	1.38
Inadequate for analysis	15	10.34
Endometrial polyp	13	8.97
Endometrial hyperplasia	45	31.03
Malignant tumors	1	0.69
Endometritis	1	0.69
Total	**145**	**100**

**Table 10 diagnostics-12-00970-t010:** Histopathological findings of clinically suitable samples extracted by scraping procedure.

Group	Case Number	Percentage (%)
Proliferative phase endometrium	23	19.83
Secretory phase endometrium	20	17.24
Menstrual phase endometrium	4	3.45
Exogenous hormones	6	5.17
Menopausal endometrium	2	1.72
Inadequate for analysis	12	10.34
Endometrial polyp	11	9.48
Endometrial hyperplasia	37	31.90
Malignant tumors	0	0.00
Endometritis	1	0.86
Total	**116**	**100**

**Table 11 diagnostics-12-00970-t011:** Histopathological findings of clinically suitable samples extracted by hysteroscopy with endometrial biopsy procedure.

Group	Case Number	Percentage (%)
Proliferative phase endometrium	9	31.03
Secretory phase endometrium	3	10.34
Menstrual phase endometrium	1	3.45
Exogenous hormones	2	6.90
Menopausal endometrium	0	0.00
Inadequate for analysis	3	10.34
Endometrial polyp	2	6.90
Endometrial hyperplasia	8	27.59
Malignant tumors	1	3.45
Endometritis	0	0.00
Total	**29**	**100**

**Table 12 diagnostics-12-00970-t012:** Suitable samples.

	D&C + HSC		D&C		HSC	
	Total Case Number	Clinically Suitable	Total Case Number	Clinically Suitable	Total Case Number	Clinically Suitable
	949	145	833	116	116	29
**Proliferative phase endometrium**						
Suitable (case number)	13		11		2	
Suitable (%)	1.37	8.97	1.32	9.48	1.72	6.90
**Secretory phase endometrium**						
Suitable (case number)	19		17		2	
Suitable (%)	2.00	13.10	2.04	14.66	1.72	6.90
**Proliferative and secretory phase endometria**						
Suitable (case number)	32		28		4	
Suitable (%)	3.37	22.07	3.36	24.14	3.45	13.79

## Data Availability

Not applicable.

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
