# Peer review of "Endometrium as Control of Endometriosis in Experimental Research: Assessment of Sample Suitability"

_diagnostics, 2022, doi:10.3390/diagnostics12040970_

Round 1

Reviewer 1 Report

None

Author Response

Thank you for your kind review. We have rephrased the manuscript with the help of a native English speaker.

Reviewer 2 Report

Comments to the authors:

The manuscript entitled “Endometrium as control of endometriosis in experimental research: assessment of sample suitability” (Manuscript ID: diagnostics-1591380) is interesting and the ability to provide information from 1162 cases may offer important information to the scientific community although does not change or improve the clinical practice directly.

The authors’ choice to perform such a study is motivating however, according to this reviewer’s opinion the manuscript is not suited for publication. The reasons are presented in specific points highlighted herein:

The introduction section provides background information about endometriosis and endometrium however it does not show the knowledge gap that this study will fill. Authors mention that (line 72-75) “it may be hypothesized that eutopic endometrium can be an appropriate control to detect the different molecular changes observed in the ectopic tissue. However, due to many reasons, it is not easy to harvest healthy human endometrium and careful consideration is needed to reveal the different molecular processes of this tissue”. Is this a hypothesis that the authors made? How does this research may contribute to the field? This is a point that merits clarification.

This is an original retrospective study however the analysis between the control and the study group is not clearly stated. Moreover, the retrospective nature of the analysis may present some limitations. The authors should discuss the possible limitations and reasons for caution when interpreting the results of the present study.

In the result section first 3 paragraphs could be introduction material and not part of the results as they show the existing difficulty in the research of endometriosis and maybe the need for a study like this. Moreover, in line 399-400 authors state that “The aim of our present retrospective analysis was to determine what proportion of the surgically removed human eutopic samples are suitable as appropriate controls to reveal molecular alterations observed in ectopic tissues.” This is a different scope of the study. What is of importance is that the authors do not provide enough molecular results in order to validate their conclusion. In general, the manuscript may provide important information to the research field however the overall flow of the present document is poor and not friendly to the readers.

Minor language corrections are needed regarding syntax and phraseology. This reviewer suggests revision of the manuscript by a native English speaker prior to publication in order to improve its performance.

Author Response

The manuscript entitled “Endometrium as control of endometriosis in experimental research: assessment of sample suitability” (Manuscript ID: diagnostics-1591380) is interesting and the ability to provide information from 1162 cases may offer important information to the scientific community although does not change or improve the clinical practice directly.

Thank you for your kind and respectful work on revision of our manuscript. We agree with the statement that the present study does not improve the clinical practice directly. Nevertheless, this research contributes to the improvement of clinical practice indirectly. Endometriosis is a chronic disease which is very hard to eliminate from the body. Nowadays, revealing alterations of molecular processes in this endometrial-like tissue is an appropriate method to provide sufficient information for pharmaceutical research institutes so that they can develop new drugs for the elimination of the ectopic tissue. To reveal which molecular processes are altered in endometriosis, it has to be compared with the physiological molecular processes observed in healthy endometrium. However, as it is detailed in the Discussion section, harvesting healthy endometrium is challenging due to the fact that the indication of surgery is generally a presupposed underlying pathological process of the endometrium. This way, if a clinician or a researcher performed experiments on endometrium, gynaecologists would use a screening algorithm before sample submission to decrease the possibility of examining molecular processes of a pathological endometrium. Thus, the aim of the present research was to determine what proportion of the surgically removed human eutopic samples are suitable as controls to reveal molecular alterations observed in ectopic tissues. This way, with the help of retrospective studies we could create the mentioned screening protocol which could be applicable in the clinical practice. Moreover, this research is useful for laboratories as well where experimental research on endometrium is performed: prior to planning the steps of the experiments, information is available for the researchers about the aspects they should consider before examining the endometrium.

Maclean, A.; Kamal, A.; Adishesh, M.; Alnafakh, R.; Tempest, N.; Hapangama, D. K., Human uterine biopsy: Research value and common pitfalls. Int J Reprod Med 2020, 2020, 9275360

The authors’ choice to perform such a study is motivating however, according to this reviewer’s opinion the manuscript is not suited for publication. The reasons are presented in specific points highlighted herein:

The introduction section provides background information about endometriosis and endometrium however it does not show the knowledge gap that this study will fill.

Thank you for your comment on the Introduction section. We rephrased the section. However, endometriosis is mentioned so many times indirectly in the manuscript, Introduction section was rewritten to avoid misunderstandings about the subject of the manuscript.

Authors mention that (line 72-75) “it may be hypothesized that eutopic endometrium can be an appropriate control to detect the different molecular changes observed in the ectopic tissue. However, due to many reasons, it is not easy to harvest healthy human endometrium and careful consideration is needed to reveal the different molecular processes of this tissue”. Is this a hypothesis that the authors made? How does this research may contribute to the field? This is a point that merits clarification.

Thank you for your question. This sentence needs explanation indeed. In Discussion section, it is detailed why ‘careful consideration is needed to reveal the different molecular processes of endometrium’ (line 433-438). It has been described in previous studies that if an experimental researcher obtains surgically removed endometrial tissue, it does not mean that the sample is healthy, or that it has been removed properly from the uterine cavity. Molecular processes of a pathological sample can be altered compared to a physiological tissue. Regarding the sentence ‘it may be hypothesized that eutopic endometrium can be an appropriate control to detect the different molecular changes observed in the ectopic tissue’, explanation of this is detailed in Discussion section as well. Cell lines and animal models are not the best choice for an experimental researcher to observe the molecular processes of endometrium. Reasons of it are detailed in lines 388-413. Endometriosis and endometrium are composed of the same stem cells and terminally differentiated cells. Moreover, the widely accepted hypothesis for the development of endometriosis is the retrograde menstruation theory. Consequently, cells of ectopic lesion migrate from the eutopic endometrium. The reason for it being hypothesised is that molecular processes of endometriosis can be compared with the physiological processes of the endometrium. Moreover, the reason for writing so many details about endometriosis (however it is not the direct subject of the manuscript) in the original Introduction section was to enable the readers of the manuscript to understand what the similarities are between the eutopic and ectopic endometrial tissues, and why they can be comparable with each other.

Sourial, S.; Tempest, N.; Hapangama, D. K., Theories on the pathogenesis of endometriosis. Int J Reprod Med 2014, 2014, 179515

McCluggage, W. G., My approach to the interpretation of endometrial biopsies and curettings. J Clin Pathol 2006, 59, (8), 801-12

Simitsidellis, I.; Gibson, D. A.; Saunders, P. T. K., Animal models of endometriosis: Replicating the aetiology and symptoms of the human disorder. Best Pract Res Clin Endocrinol Metab 2018, 32, (3), 257-269

Fan, H., In-vitro models of human endometriosis. Exp Ther Med 2020, 19, (3), 1617-1625

This is an original retrospective study however the analysis between the control and the study group is not clearly stated. Moreover, the retrospective nature of the analysis may present some limitations. The authors should discuss the possible limitations and reasons for caution when interpreting the results of the present study.

Thank you for your kind advice regarding the limitations of the study. We added some information to the Discussion section about the possible limitations and reasons for caution when interpreting the results of the present study.

However, in this retrospective analysis there were no control and study groups. The type of research differed from this, so no groups were compared to each other. We only collected data from 1162 cases. These cases were grouped, case numbers and percentages were calculated. Then, consequences were drawn from these data. Regarding the 8 cultured samples, there were control and treated groups. Differences between the two were detailed in Results section (line 296-373).

In the result section first 3 paragraphs could be introduction material and not part of the results as they show the existing difficulty in the research of endometriosis and maybe the need for a study like this. Moreover, in line 399-400 authors state that “The aim of our present retrospective analysis was to determine what proportion of the surgically removed human eutopic samples are suitable as appropriate controls to reveal molecular alterations observed in ectopic tissues.” This is a different scope of the study. What is of importance is that the authors do not provide enough molecular results in order to validate their conclusion.

The first 3 paragraphs of Result section are part of the results as data from the retrospective analysis are detailed in these paragraphs. Moreover, in Results section there is no data about endometriosis samples. The 3 paragraphs only detail what the numbers and percentages are of the different diagnoses described from the 1162 eutopic endometria harvested with D&C and HSC biopsy procedures.

At first glance, it is a logical thought why the conclusion is not validated with molecular results. However, aim of the present research was not to describe molecular alterations between physiological and pathological endometria. It is detailed in the Discussion section (line 452-458) that other researchers observed altered molecular processes between the two endometria. The aim of the present study was to provide information for the research community about the possibility of getting healthy, surgically removed endometrium. If someone performs experiments with endometriosis, results obtained from this tissue have to be compared with the result of physiological endometrium. Moreover, parallelly to this present research, our laboratory has an experimental research on endometriosis and endometrium as well. We could detect differences between physiological and pathological endometria. Furthermore, differences were observed between the molecular processes of endometrium and endometriosis, e.g. administration of different drugs can change mRNA expression of VEGF (Figure). We have not described these results in the present study, because these are preliminary results and we would like to publish it in a future manuscript. Therefore, we would not like to publish it as a supplementary information. Moreover, the aim of the present study was not to show molecular alterations between the different endometrial tissues.

In general, the manuscript may provide important information to the research field however the overall flow of the present document is poor and not friendly to the readers.

Thank you for considering the present manuscript important for the research field. It is needed to be clarified why it is not poor for the research community. The present manuscript is more useful for researchers who do experimental research on endometrium and endometriosis. Before planning their experiments, they need sufficient information about this subject and the pitfalls they could potentially face during the research. Working with a tissue like endometrium is challenging and being aware of the possibility of getting surgically removed physiological endometrium is essential. In previous manuscripts this all had not been described yet. Furthermore, this manuscript is also useful for clinicians (gynaecologists) who collaborate on endometrial tissues with experimental researchers. Harvesting endometrium from the uterine cavity is an available and easily achievable process. But for an in vitro research, only physiological endometrium is suitable to be submitted by the surgeons for further experimental studies if endometriosis is needed to be compared with endometrium. The described steps of screening protocol help the gynaecologists to decrease the chance of submitting pathological endometrium to a laboratory that does in vitro experiments on this sample.

Minor language corrections are needed regarding syntax and phraseology. This reviewer suggests revision of the manuscript by a native English speaker prior to publication in order to improve its performance.

Thank you for your kind suggestion. Manuscript was revised and corrected by a native English speaker.

Reviewer 3 Report

The study tries to tackle the problem for the eutopic endometrium evaluation models. The use of samples collected previously create a difficulty in terms of number of valid samples, the authors ending up with 32 samples from 949 at origin, which create a weakness for the design of the study.

The abstract could be more structured

The material and method section is exhaustively explanatory, but a bit difficult to follow. The diagram at the end completed with numbers could help.

The results section contains also individual cases with photos, which could be submitted separately as supplementary material for the ease of reading. Also, the exclusion of endometriosis cases is not very clear- regarding on what criteria- as this also could influence the result. 

Overall, although the idea of the study is interesting, the design and the readibility can be improved.

Author Response

Thank you for your kind revision of our manuscript.

Changes were highlighted in the manuscript.

The abstract was structured as you kindly suggested.

Figures in the ‘Results’ section were submitted separately as supplementary figures.

Regarding your question about endometriosis cases, these samples were excluded, because in patients where the ectopic tissue develops, intrauterine endometrium is capable for the formation of the ectopic tissue. Thus, these samples are not considered to be healthy endometria.

The manuscript was supervised and corrected by a native English speaker.

Round 2

Reviewer 2 Report

Comments to the authors:

I would like to thank you for providing your revised manuscript entitled “Endometrium as control of endometriosis in experimental research: assessment of sample suitability” (Manuscript ID: diagnostics-1591380) addressing the points highlighted by the reviewer.

According to this reviewer's opinion, authors present a revised manuscript addressing the points highlighted by this reviewer and the revised manuscript meet the high standards for publication in “Diagnostics”. Therefore, the recommendation is “Accept”.

Author Response

Thank you for your kind revision of our manuscript.

Changes were highlighted in the manuscript.